# Does Long-Term Care Literacy Matter in Evaluating Older Care Recipients’ Satisfaction with Care Managers? Empirical Evidence from Japanese Survey Data

**DOI:** 10.3390/ijerph20032456

**Published:** 2023-01-30

**Authors:** Ziyan Wang, Kaori Fukayama, Bing Niu

**Affiliations:** 1Graduate School of Economics, Osaka Metropolitan University, Osaka 599-8531, Japan; 2Graduate School of Nursing, Osaka Metropolitan University, Osaka 553-8555, Japan

**Keywords:** care literacy (CL), care manager (CM), long-term care insurance (LTCI), satisfaction

## Abstract

In this study, we focused on the long-term care literacy of care recipients (older adults currently receiving formal care services) and examined its relationship with satisfaction with their care managers by using a unique individual dataset of Japanese people aged 65 years and older. To address the problem of non-respondent bias, we applied inverse probability weighting and the Heckman probit model for estimation. We found that the probability of older adults evaluating their satisfaction regarding the six aspects of care manager measurement increased with an increasing level of care literacy. However, concerning the level of satisfaction with their care managers, we only observed significant increases in the aspects of “Explanation power” and “Attitude and manners” as the level of care literacy increased. Covariates, such as age, gender, family structure, level of certification for long-term care, reasons for choosing the care manager, utilization of long-term care services, and the manner in which older respondents answered the survey questions, also mattered regarding the evaluation process of satisfaction of older adults. In Japan, utilizing formal care services based on the long-term care insurance system is complicated and sometimes difficult for older adults to understand. In this survey, 35% of older care recipients had inadequate care literacy. Improving the care literacy of older adults is important for better use of formal care services and increased satisfaction.

## 1. Introduction

Long-term care literacy (CL) is the ability of citizens who are not engaged in the long-term care business to obtain basic knowledge about the long-term care insurance (LTCI) system and caregiving. It entails the proper use of knowledge in understanding the problems faced by older adults, care recipients, and patients with dementia [1]. CL does not refer to the ability to individually care for someone but to have insight into the long-term care services available within society. Japan’s current LTCI system and private services are complex and diverse, and understanding and using them is indispensable.

In Japan, older adults acquire health and medical information mainly from television, newspapers, local publications, and word of mouth from friends or family [2,3]. However, in the process of obtaining information, approximately 15% of older adults are dissatisfied because they do not know where to obtain reliable information [4]. According to a survey on the daily lives and participation in local communities of older adults in 2021, approximately 30% of older adults answered that they needed more information on health promotion, and 35% answered that they needed more information about health care and nursing care services such as home care [4]. This lack of information can lead to low CL among older adults. Individuals with inadequate CL may find it difficult to make appropriate decisions regarding long-term care services and may ignore expert advice [1]. Care recipients and families with inadequate CL may find it challenging to fully understand the content of the services provided. In some cases, they are dissatisfied because they misunderstand the meaning of the correspondence provided by doctors and care managers (CMs) [1].

In Japan, the ratio of the older population (aged 65 and over) to the total population has been consistently increasing since 1995 (5.3%), exceeding 10% in 1985, 20% in 2005, and reaching 29.1% in 2022 (Figure 1) [5]. Japan is facing a “population and social security problem” where the population aged 75 and over is expected to grow to 21.8 million by 2025; therefore, it is predicted that one in five people will be aged 75 and over. In a technical paper on how the super-aging society has progressed, the Ministry of Health, Labour and Welfare (MHLW) cites several problems regarding the anticipated social composition of 2025 [6]. First, regarding the transition of the population of older adults, the “high” aging rate has become a problem because of the “speed” of the aging process. Second, the number of older adults with dementia has reached approximately 3.2 million and is expected to increase rapidly hereafter. Third, the number of households with older adults has reached approximately 18.4 million, and about 70% are either living alone or with a spouse only. Finally, with such an increase in the population of older adults, the needs and costs of their health care and long-term care are also expected to increase. Since 2000, the percentage of people with a certified requirement for long-term care among the population aged 65 and over has been increasing consistently, as shown in Figure 2. In 2020, 18.7% of older adults were certified for long-term care, and 6.4% had severe nursing care needs (“Long-term care” levels 3–5).

In anticipation of an increasing number of older adults who will need nursing care in the future along with the aging population, the introduction of a care management system under long-term care insurance (LTCI) in 2000 shifted authority from the physician to the CM [9]. Under this system, a CM plays an important role in connecting the demand and supply of long-term care services. Anyone who needs long-term care and wishes to receive formal care services must be certified as needing long-term care and consult their needs with a CM, which can be introduced by public institutions, referred by word of mouth, or found by themselves. The CM creates a care plan, and care services are started based on that plan. Specifically, CMs help older clients, families, and other caregivers by: (1) conducting care-planning assessments to identify needs, problems, and eligibility for assistance, (2) screening, coordinating, and monitoring home care services, (3) providing client and family education and advocacy, and (4) offering counseling and support [10]. CMs also serve as an important source of medical information for older adults, their families, and their communities [10]. Figure 3 and Figure 4 show specific workflows of CMs regarding in-home nursing care and services offered at facilities [11].

In recent years, there has been an increase in the number of CMs who become licensed caregivers but lack the skills to either communicate effectively with physicians and nurses or handle medical problems [10]. According to the MHLW, the following four conditions must be met to become a CM: (1) have more than five years of practical experience in the field of health care and welfare (doctors, nurses, social workers, care workers, etc.), (2) pass the practical training examination for long-term care support specialists (CMs), (3) complete the practical training course, and (4) have received a certificate for becoming a CM [11].

In Japan, CMs have different basic qualifications and backgrounds. The basic qualifications can be broadly divided into care workers, social workers, and nurses, and the practical content differs depending on those basic qualifications. However, in reality, care and social workers have the most qualifications for CMs, and the total of both accounts for more than 80% of CMs [12]. Differences in the practice of care management for older care recipients, owing to basic qualifications [12,13,14,15], are natural because of the differences in original education and expertise; however, they pose a problem in maintaining the quality of care management, which may affect the quality of care provided to their clients [16,17].

Owing to the few empirical studies on CMs in Japan, it is important to research client satisfaction. Kuroda et al. [18] found that caregivers’ mental quality of life (QOL) was significantly related to their sense of coherence and satisfaction with their CM. Jubelt et al. [19] found that patients’ favorable perceptions of case managers were associated with a higher overall satisfaction with care and may lower the risk of future acute care.

This study focuses on the CL of care recipients (older adults currently receiving formal care services) and examines its relationship with satisfaction with CMs using a unique individual dataset of the Japanese population aged 65 years and older. Few studies have examined the impact of CL on care outcomes. However, a qualitative study by Saito [1] showed that inadequate CL negatively impacted out-of-pocket costs and satisfaction with formal care. We hypothesize the path of the impact of CL on satisfaction with CMs as follows: Care recipients with adequate CL understand the presented care services that meet their real needs better than care recipients without. They can more easily express and request the services they want and might receive a good care plan that results in higher satisfaction. We further propose that adequate CL may contribute to a better understanding of the complicated long-term care system, utilization of its care services, and the CM’s response. These aspects may increase the degree of satisfaction with CMs.

## 2. Methods

### 2.1. Data

We applied a data set obtained from a 2019 survey conducted by Sakai City regarding the living conditions, health, and welfare of older adults in Osaka Prefecture, Japan [20]. A sample of 2000 respondents aged 65 and over who live in the city and are certified with different long-term care levels ranging from the mildest level of “Requiring help 1” to the most severe level of “Long-term care level 5” were randomly selected for the survey. Permission to use the data was granted after applying for information disclosure with the Health and Welfare Bureau of Sakai City.

There are several advantages to using this dataset in our study. First, it is a unique dataset encompassing various characteristics and detailed living conditions of older adults. It contains variables such as age, gender, household structure, economic status, community activities, family doctor status, utilization of long-term care services, and required care level. The dataset also includes information on body mass index (BMI), instrumental activities of daily living (IADL), subjective health status, and smoking status, among other variables. These covariates are important for analyzing the decision-making of older adults and are, therefore, included in the estimation framework. Second, the data include related question items needed to develop an index for CL. To evaluate CM satisfaction from different dimensions, question items from various aspects are available. Third, this dataset is sufficiently representative, as the demographic and socio-economic characteristics (level of income, education, and academic ability) of Sakai City were consistent with those of the whole country at the time of the study. A comparison of these characteristics between Sakai City and the entire country of Japan is presented in Figure 5.

### 2.2. Measurement of CL

In this study, we generated an original index for CL based on the following three aspects: (1) knowledge about LTCI and dementia, (2) prior knowledge about how to choose a nursing care facility and its admission, and (3) prior knowledge required from long-term care service providers for their users [1]. We selected the survey items from data corresponding to the above aspects. First, we considered the question related to LTCI (i.e., “Which of the following is closest to your opinion regarding future LTCI premiums in Sakai City?”). As for the choices, four options regarding opinions that included “not sure” were presented to the respondents. Second, regarding nursing care facilities, respondents were asked about the degree of recognition (either knowing or not knowing) of the three types of services (i.e., “small-scale multifunctional home care”, “nursing small-scale multifunctional home care [combined service]”, and “regular patrol/on-demand home-visit nursing care”). Finally, regarding the future needs of care, we asked, “What kind of long-term care do you want in the future?” Seven choices, including the “not sure” option, were presented to the respondents.

Based on the responses to the above questions, we generated a comprehensive index for CL in two steps. In step one, each aspect of (1) LTCI, (2) nursing care facility, and (3) future needs of care were indexed as follows: For (1) and (3), we set the index to 1 if the respondent had sufficient knowledge of LTCI and future needs of care (when the respondent chose an option with a detailed opinion) and 0 if the respondent had insufficient knowledge (when the respondent answered “not sure” or did not answer). For (2), regarding each of the three types of nursing care services, we set the index to 1 if the respondent knew it and 0 if not. If the sum of values (0 to 3 points) of the three questions was 0 to 1 point, we assumed that the respondent had insufficient knowledge and set it to 0. If the sum of values was 2 to 3 points, we assumed that the respondent had sufficient knowledge and set it to 1.

In step two, based on the dummy variables of the above three aspects and the sum of values (0 to 3 points), we set the comprehensive index for CL to 0 if the sum of values was 0 to 1 point, indicating that the respondent had “inadequate CL”. We set the index to 1 if the sum of values was 2 to 3 points, indicating the respondent had “adequate CL”. The details of the index setting based on these three aspects are presented in Table 1. Figure 6 shows the distribution of CL among the current long-term care recipients. In our study, 35% of older care recipients had inadequate CL.

### 2.3. Measurement of Satisfaction with CM

We evaluated the satisfaction of care recipients with their CM from six different aspects, as follows: (1) Expertise was captured by asking the question, “Does the CM have a wide range of knowledge about long-term care and health care, and can he/she give professional advice when creating a care plan?”; (2) Information provision was captured by asking, “Does the CM provide you information on various care services and service providers?”; (3) Explanatory power was captured by asking about “clarity of the care plan description”; (4) Communication was captured by asking, “Does the CM talk to the care provider about issues that you find difficult to communicate?”; (5) Responsiveness was captured by asking, “Does the CM respond when you want to talk or need an urgent response?”; and (6) Attitude and manners were captured by asking about the “attitudes and manners such as keeping time”. Respondents were presented with the following choices to each question: “Satisfied”, “Satisfied, if I have to choose”, “Dissatisfied, if I have to choose”, “Dissatisfied”, and “Not sure”.

The behavior of care recipients’ evaluation of satisfaction with CMs was analyzed in two stages in this study. In the first stage, we focused on the need to evaluate one’s CM. We adopted methods from previous studies on satisfaction [27] and assigned a dummy variable representing the behavior of the care recipient’s evaluation. We set it to 0 if there was no evaluation (respondent either answered “Not sure” or gave no response) and 1 if there was an evaluation (respondent either answered “Satisfied”, “Satisfied, if I have to choose”, “Dissatisfied, if I have to choose”, or “Dissatisfied”). In the second stage, we focused on evaluating the level of satisfaction. We generated another dummy variable that represents the degree of satisfaction and set it to 0 if it was evaluated as either “Dissatisfied” or “Dissatisfied, if I have to choose”, and 1 if it was evaluated as either “Satisfied” or “Satisfied if I have to choose”. The setting of the dummy variables for both stages is presented in Table 2.

### 2.4. Covariates

Based on a review of some previous studies on health literacy [28,29,30,31,32], we controlled the following variables as covariates in the regression analysis: demographic variables (age (ranges 65–69, 70–74, 75–79, 80–84, 85–89, and ≥90), gender, family structure (living alone, couple only, or others)), and certified degree of long-term care required (“Requiring help” levels 1–2, “Long-term care” levels 1–2, and “Long-term care” levels 3–5). We also controlled for dummy variables such as the one representing the reason for choosing the CM’s office (introduced or other reasons), the variable representing the utilization status of long-term care services (yes/no), and the variable representing the manner in which the care recipient answered the question (answered with the assistance of family members/relatives and/or others present or answered alone).

### 2.5. Estimation Framework

The study sought to understand the degree of satisfaction with CMs. Approximately 20–30% answered “Not sure” and did not answer/evaluate the question on CMs. The distribution of responses to satisfaction with CMs regarding the six aspects of CM measurement is presented in Table 3. The high percentage of those who did not evaluate clearly can be attributed to sampling bias. It was necessary, therefore, to reconsider the analytical design to correspond with the problem. We used the following two approaches to address the problem.

#### 2.5.1. Inverse Probability Weighting

First, we used propensity weighting, a widely accepted technique to adjust for non-respondent bias which is applied broadly in healthcare empirical studies [32,33].

We modeled the probability of response/evaluation as a function of CL. Covariates that include demographic variables (age, gender, and family structure), variables related to the use of long-term care services (certified degree of long-term care required, reason for choosing the CM’s office, utilization status of long-term care services), and a variable representing the manner in which the care recipient responded to the question were used. We then used the estimated probability to generate an inverse probability of treatment weight (IPTW). This was applied as a weighted variable to the estimation of the satisfaction level, making those who responded better resemble all the targets (care recipients) who were part of the survey. The calculation Formula is as follows:(1)$$Probit\left(\hat{Z}\right)=XB+\epsilon ,$$
where *X* is a covariate vector and *B* is a vector of coefficients.

We then calculated inverse probability weights and applied them to estimate the satisfaction level, that is
(2)$$Weight=\frac{T}{P}+\frac{1-T}{1-P},$$
where *T* is a binary treatment and *P* is the obtained inverse probability.

#### 2.5.2. Bivariate Probit Model with Sample Selection

Second, we used the bivariate probit model with sample selection to handle the problem of non-respondent bias [34,35,36]. This approach, also known as the Heckman probit model, performs econometric analysis in two stages. Here, in the first stage, the evaluation was performed explicitly, and the data were collected regarding satisfaction level. These data served as selected samples for the second stage.

In the initial stage, we set up a probit model of the selection equation to determine whether a care recipient makes a clear evaluation. In the next stage, we set up another probit model of the outcome equation with a dichotomous level of satisfaction with a CM (0 = dissatisfied or 1 = satisfied) with respect to the selected samples obtained from the previous stage. The settings of the equations in both stages are as follows:

Selection (whether or not to evaluate) Equation:(3)$$z_{i}^{*}=w_{i}\gamma +v_{i}$$
(4)$$z_{i}=\left\{\begin{array}{c} \begin{array}{cc} 1 & if\;z_{i}^{*}>0 \end{array}\\ \begin{array}{cc} 0 & if\;z_{i}^{*}\leq 0 \end{array} \end{array}\right.$$

Outcome (satisfaction level) Equation:(5)$$y_{i}=\left\{\begin{array}{c} \begin{array}{cc} x_{i}\beta +u_{i} & \;if\;z_{i}^{*}>0 \end{array}\\ \begin{array}{cc} unobservable & if\;z_{i}^{*}\leq 0 \end{array} \end{array}\right.$$
where the assumptions of the error terms in both Equations are as follows.
(6)$$\begin{array}{c} v_{i}\sim N\left(0,\;1\right)\\ u_{i}\sim N\left(0,\sigma^{2}\right)\\ \mathrm{corr}\left(v_{i},u_{i}\right)=\rho \end{array}$$

Because problems arise when estimating β if $v_{i}$ and $u_{i}$ are correlated $\left(\rho \neq 0\right)$, the standard probit model applied to the outcome equation yields biased results. To obtain consistent, asymptotically efficient estimates for all parameters in our model, we used the Heckman probit model. If we failed to reject the null hypothesis that ρ=0 based on the likelihood ratio test of independent equations, we then further applied the standard probit model to estimate the outcome equation.

The three types of observations in our sample had probabilities that are defined as follows:(7)$$z_{i}=0\;\;\;Pr\left(z_{i}=0\right)=\Phi \left(-w_{i}\gamma \right)=1-\Phi \left(w_{i}\gamma \right)$$
(8)$$z_{i}=1,\;y_{i}=0\;\;\;Pr\left(z_{i}=1,\;y_{i}=0\right)=\Phi_{2}\left(w_{i}\gamma ,-x_{i}\beta ,-\rho \right)$$
(9)$$z_{i}=1,\;y_{i}=1\;\;\;Pr\left(z_{i}=1,y_{i}=1\right)=\Phi_{2}\left(w_{i}\gamma ,x_{i}\beta ,\rho \right)$$

The log-likelihood function is
(10)$$\;\;\;\;lnL=\sum_{i\notin S}ln\left\{1-\Phi \left(w_{i}\gamma \right)\right\}+\sum_{\begin{array}{c} i\in S\\ y_{i}=0 \end{array}}ln\left\{\Phi_{2}\left(w_{i}\gamma ,-x_{i}\beta ,-\rho \right)\right\}+\sum_{\begin{array}{c} i\in S\\ y_{i}=1 \end{array}}ln\left\{\Phi_{2}\left(w_{i}\gamma ,x_{i}\beta ,\rho \right)\right\}$$
where S is the set of observations for which $y_{i}$ is observed, $\Phi \left(\cdot \right)$ is the standard cumulative normal distribution function, and $\Phi_{2}\;\left(\cdot \right)$ is the cumulative bivariate normal distribution function.

## 3. Results

### 3.1. Descriptive Statistics

The descriptive statistics of the main variables in this study are summarized in Table 4. The dependent variables are summarized for each stage of the analysis. Among our observations, 64.7% of the older adults had adequate CL. The majority of respondents were female (71.0%), older adults aged 75 years and over (84.2%), and either living alone (39.7%) or living with a spouse only (36.1%). Regarding the degree of care, 66.1% of the respondents were certified at low levels (“Requiring help” levels 1–2). Nearly half of the respondents chose their CMs based on referrals from others (45.9%). The utilization of long-term care services was 54.8%. Of the respondents, 15% answered the questionnaire with the assistance of their family members, relatives, or others present.

### 3.2. Results of Inverse Probability Weighting

The estimated marginal effects of CL and other covariates on satisfaction with CMs via the two stages are summarized in Table 5.

We found that the probability of an older adult making a clear evaluation of all six aspects of CM measurement would increase significantly by 6.5% to 14.1% if they had adequate CL. Older adults aged 65 to 69 years were significantly more likely to make a clear evaluation only towards “Responsiveness” by 17.2%, but older adults aged 75 to 79 years were more likely to respond to other aspects such as “Information provision”, “Responsiveness”, and “Attitude and manners” by 10.5%, 10.8%, and 7.4%, respectively. Compared to the low-level group of a certified degree of long-term care required (“Requiring help” levels 1–2), the middle-level group (“Long-term care” levels 1–2) was significantly more likely to make a clear evaluation towards “Communication” and “Responsiveness” by 10.1% and 10.4%, respectively, while the high-level group (“Long-term care” levels 3–5) was more likely to respond to more aspects such as “Explanation power”, “Communication”, and “Responsiveness” by 12.3%, 18.8%, and 10.7%, respectively. Except for “Explanation power”, the probability of making a clear evaluation of the five aspects increased significantly by 5.8% to 13.0% when older adults were using long-term care services. Moreover, if older respondents answered the survey with the assistance of others (family/relatives/others), they were significantly more likely to make a clear evaluation towards the aspects of “Expertise”, “Explanation power”, “Responsiveness”, and “Attitude and manners” by 8.2%, 11.9%, 9.1%, and 10.0%, respectively.

In the second stage of satisfaction level, we found that if older adults had adequate CL, the probability of making an evaluation as being satisfied would increase significantly in the aspects of “Explanation power” and “Attitude and manners” by 5.0% and 3.7%, respectively. For older adults aged 85 to 89 years, the probability of being satisfied with “Explanation power” would increase significantly by 5.4%. Compared to women, men were 4.4% and 4.5% more likely to rate themselves as being satisfied with “Expertise” and “Explanation power”, respectively, but 7.0% lower with “Responsiveness”. In the case of older adults living alone, satisfaction with “Responsiveness” decreased significantly by 13.9%. Compared to the low-level group of care (“Requiring help” levels 1–2), satisfaction with “Responsiveness” of the middle-level group (“Long-term care” levels 1–2) increased significantly by 6.5%. In the cases where others referred a CM, the probability of being satisfied decreased significantly by 5.6%, 5.5%, 5.9%, and 5.6% towards the four aspects of “Expertise”, “Information provision”, “Explanation power” and “Communication”, respectively.

### 3.3. Results of the Heckman Probit Model

The estimated marginal effects of CL and other covariates from both the selection and outcome equations are summarized in Table 6. The sign, magnitude, and significance of the estimates were consistent with the results of the inverse probability weighting.

The probability of older adults evaluating all six aspects of CM measurement would increase significantly from 6.5% to 14.1% if they had adequate CL, as revealed by the selection equation output. The effects of different age groups are similar in magnitude and significance to the results of the inverse probability weighting approach. The higher the level of certification for long-term care, the higher the probability of making an explicit evaluation, especially in terms of “Explanation power”, “Communication”, and “Responsiveness”. For those who were utilizing long-term care services, the probability of evaluating the aspects of “Expertise”, “Information provision”, “Communication”, and “Responsiveness” increased significantly. For those older adults responding to the survey questions with the assistance of others, the probability of evaluating their satisfaction with most aspects of CM measurement also increased significantly.

When the outcome equation was applied, we found that the probability of older adults evaluating themselves as being satisfied with “Explanation power” would increase significantly by 4.9% if they had adequate CL. Older adults aged 85 to 89 years were more likely to be satisfied with “Explanation power” by 4.9%. Compared to women, men were 4.2% and 4.6% more likely to rate satisfaction with “Expertise” and “Explanation power”, respectively. When living alone, satisfaction with “Responsiveness” decreased significantly by 7.8%. Where CMs were referred by others, the probabilities of being satisfied decreased significantly by 5.6%, 5.1%, and 5.5% towards “Expertise”, “Information provision”, and “Explanation power”, respectively. Responding with someone else present increased the satisfaction level by less than 1%.

After estimation, we performed the likelihood ratio test for independent equations on each aspect, and we did not reject the null hypothesis that the equations are independent (ρ = 0) based on the test results. Therefore, we further applied the standard probit model to estimate the outcome equation independently and compare the results with previous models.

### 3.4. Results of the Probit Model

The estimated marginal effects are summarized in Table 7. In the standard probit model, the results of the first stage were the same as those from the inverse probability weighting approach. We, therefore, focused on the results of the second stage and compared them with those from the Heckman probit model.

Older adults with adequate CL were more likely to make an evaluation of being satisfied with “Explanation power” and “Attitude and manners”. The results were statistically significant, with probabilities of 5.0% and 3.8%, respectively. Only the estimates for “Explanation power” were almost the same in both the Heckman probit model and the probit model.

Regarding other covariates, both the Heckman probit model and the probit model produced similar results for the sign, magnitude, and significance of the estimates in the second stage.

## 4. Discussion

Based on the results of this study, several observations emerged: First, care recipients with adequate CL were significantly more likely to make a clear evaluation regarding all six aspects of CM measurement, and we observed consistent results for all models. Care recipients with adequate CL may understand the long-term care system, the presented care services that meet their actual needs, and the response of CMs better than care recipients without. These aspects may increase satisfaction with CMs, and we further observed that those with adequate CL were significantly more likely to evaluate the “Explanation power” and “Attitude and manners” of CMs as satisfactory. Older adults are sometimes conservative and tenacious and, therefore, less adaptable to new environments; they want to be respected for their own experiences [37]. CMs visit the place where the care recipient is accustomed to living and adjust social resources to realize the life desired by the care recipient. Therefore, care recipients with adequate CL demand that their lifestyles and values be respected regarding the services and care plans they receive. The politer “Explanation power” and “Attitude and manners” of CMs might be required and more highly evaluated by older care recipients.

Second, in Japan, older adults are divided into those in the early stage of life (aged 65 to 74 years) and those in the latter stage of life (age 75 years and over), and there are differences between them in terms of their needs for support and long-term care and the status of the actual use of formal care services. Therefore, the point of focus of the evaluation differs depending on the specific age group; for example, those aged 65 to 69 years evaluated “Responsiveness” more, while those aged 75 to 79 years evaluated more various aspects such as “Information provision”, “Responsiveness”, and “Attitude and manners”. To increase the satisfaction of care recipients, CMs need to respond with awareness to the needs of different age groups and the evaluation points they place particular importance on.

Third, men were significantly more likely to evaluate the “Expertise” and “Explanation power” of CMs as satisfactory. Since some believe men to be more realistic and research-oriented than women [38], they may value the “Expertise” and “Explanation power” of their CMs regarding their continued living at home with health concerns.

Fourth, when CMs were referred by others, the probability of care recipients being satisfied with the aspects of “Expertise”, “Information provision”, and “Explanation power” decreased significantly. Moreover, older adults living alone were significantly less likely to evaluate their satisfaction with the “responsiveness” of CMs. Many older adults living alone tend to have poor eyesight, forgetfulness, and depression, even if they are independent in their basic ADL. Furthermore, some have no sense of purpose in living after the death of their spouse or family member(s) [39]. Highly self-reliant older adults with roles and strong social networks have a purpose in life [40]. Care recipients who were referred to a CM by another person were found to be less satisfied with “Expertise”, “Information provision”, and “Explanation power” because they might have lower needs for their roles and social networks and have less autonomy than those who referred their CM. In addition, older adults living alone might have less autonomy and are likely to ask others for help. Therefore, they are more likely to be less satisfied with the “responsiveness” of their CM.

The study has several limitations. First, when estimating satisfaction with CMs, the attributes and skills of the CM, such as years of experience, the number of care plans created per month (workload), and affiliation, should be controlled in the analysis. However, this dataset mainly focused on the attributes of care recipients on the demand side, and there was little information on the characteristics of CMs on the supply side of long-term care services. Second, the CL itself may change depending on the period of use of the long-term care service or the duration/opportunities for contact with the CM. In this analysis, the data is limited to the current utilization status of long-term care services; other specific usage situations of the service should also be considered in future analyses.

## 5. Conclusions

In this study, we focused on the CL of older care recipients and examined its relationship with satisfaction with their CMs using a dataset of older Japanese adults. The results show that the probability of the older adults evaluating their satisfaction regarding six aspects of CM measurement increased with increasing levels of CL. However, concerning the level of satisfaction with CMs, only the aspects of “Explanation power” and “Attitude and manners” significantly increased with increasing levels of CL. Covariates such as age, gender, family structure, level of certification for long-term care, reasons for choosing the CM, utilization of long-term care services, and the manner in which older respondents answered the survey questions also mattered regarding the evaluation process of satisfaction among older adults.

The purpose and philosophy of the LTCI Law of Japan are to provide the necessary support so that older adults can maintain their dignity and lead an independent daily life according to their abilities, even if they require long-term care [41]. However, utilizing services based on the LTCI system is complicated and may be difficult for ordinary older adults to understand. In Japan, many older adults have reported needing more reliable information about health promotion, health care, and nursing care [4]. Obtaining sufficient information may lead to improved literacy and improving the CL of older adults is important for better use of formal care services and increased satisfaction.

In Japan, it is reported that 74% of people want to receive long-term care at home when the need arises [41]. It is an important policy issue to provide a comprehensive community care system that enables older adults to continue living with dignity in the area they are accustomed to, even if they become long-term care recipients [41]. Therefore, CMs will be required to comprehensively evaluate appropriate social resources and provide them to care recipients in the future. Against the backdrop of the growth in the need for nursing care along with the aging population in the future, it is also important to improve both the qualifications of CMs and the quality of care management [42] to increase the satisfaction of their care recipients.

## Figures and Tables

**Figure 1 ijerph-20-02456-f001:**
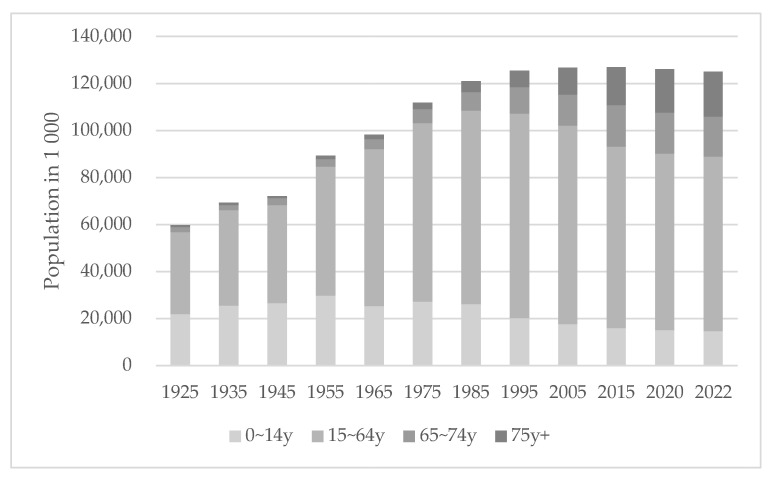
Population structure by age group in Japan (1925–2022). Data source: Ministry of Internal Affairs and Communications, Population Estimates (2022) [7].

**Figure 2 ijerph-20-02456-f002:**
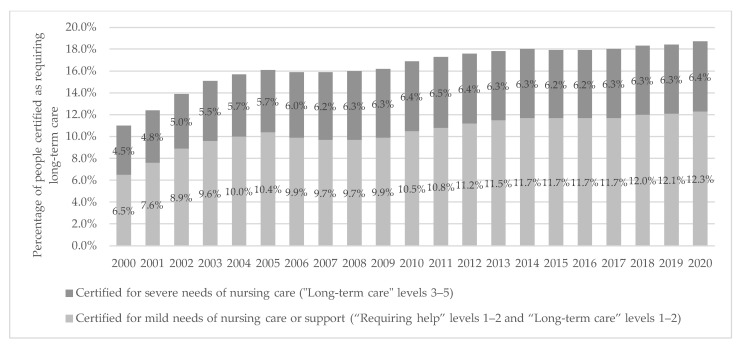
Percentage of people certified as requiring long-term care among older adults (2000–2020). Data source: Ministry of Health Labour and Welfare, Status report on Long-term Care Insurance (2020) [8]. Note: “Requiring help” and “Long-term care” levels 1–2 were included in the mild needs of nursing care from 2000 to 2005.

**Figure 3 ijerph-20-02456-f003:**
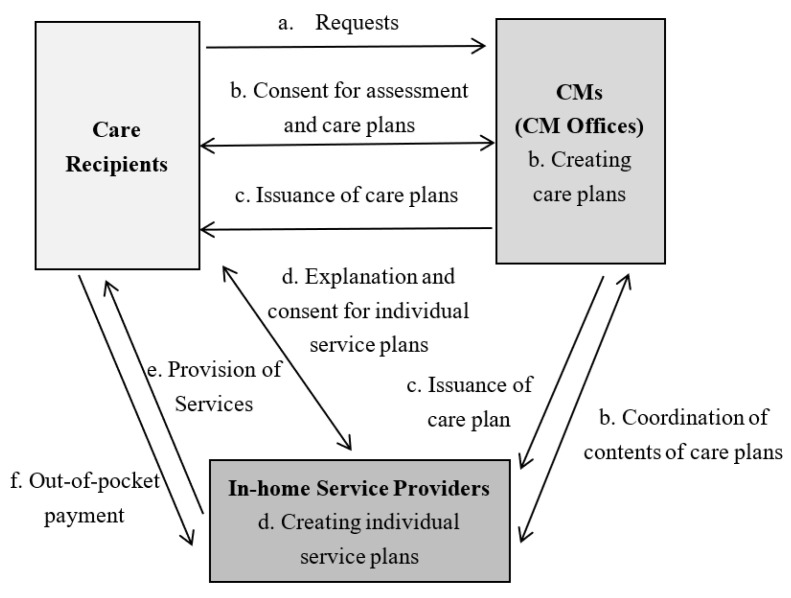
Workflow of CMs regarding in-home nursing care. Source: Ministry of Health, Labour and Welfare. Instructions for Care Managers, 2023 [11].

**Figure 4 ijerph-20-02456-f004:**
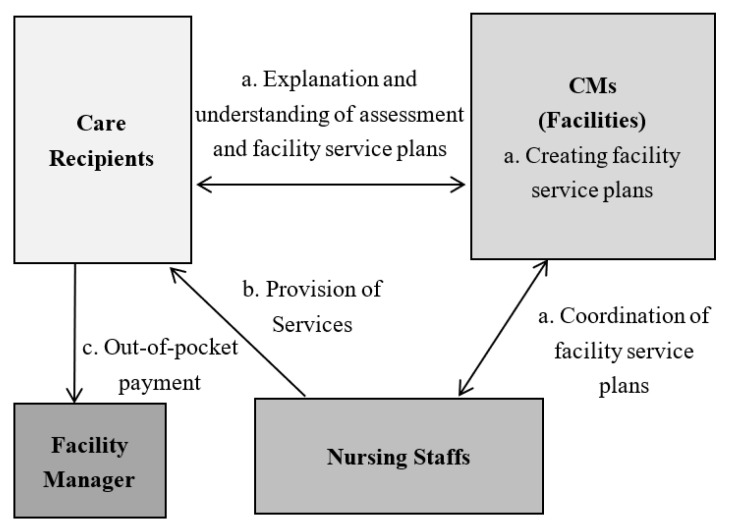
Workflow of CMs regarding services offered used at facilities. Source: Ministry of Health, Labour and Welfare. Instructions for Care Managers, 2023 [11].

**Figure 5 ijerph-20-02456-f005:**
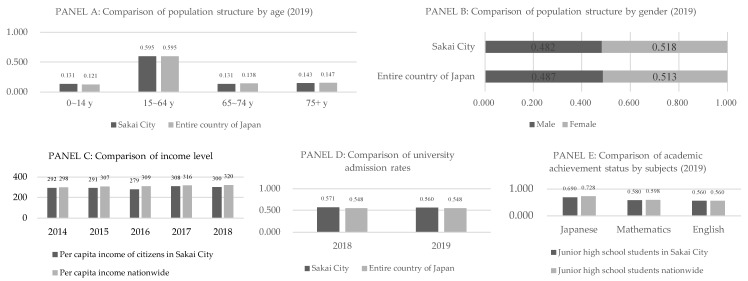
A comparison of Sakai City and the country as a whole in terms of population composition, income, and education level. Data source: Panels (**A**,**B**): Sakai City, Basic Resident Register (including foreign residents, by age and area) (2019) [21]; Statistics Bureau of Japan, Annual Report of Population Estimates (2019) [22]. Panel (**C**): Sakai City, Economic Calculation of Sakai Citizen (2018) [23]; Cabinet Office, Annual Report on National Accounts (2018) [24]. Panel (**D**): Ministry of Education Culture Sports Science and Technology of Japan, School Basic Survey (2019) [25]. Panel (**E**): National Institute for Educational Policy Research of Japan, National Assessment of Academic Ability Survey (2019) [26].

**Figure 6 ijerph-20-02456-f006:**
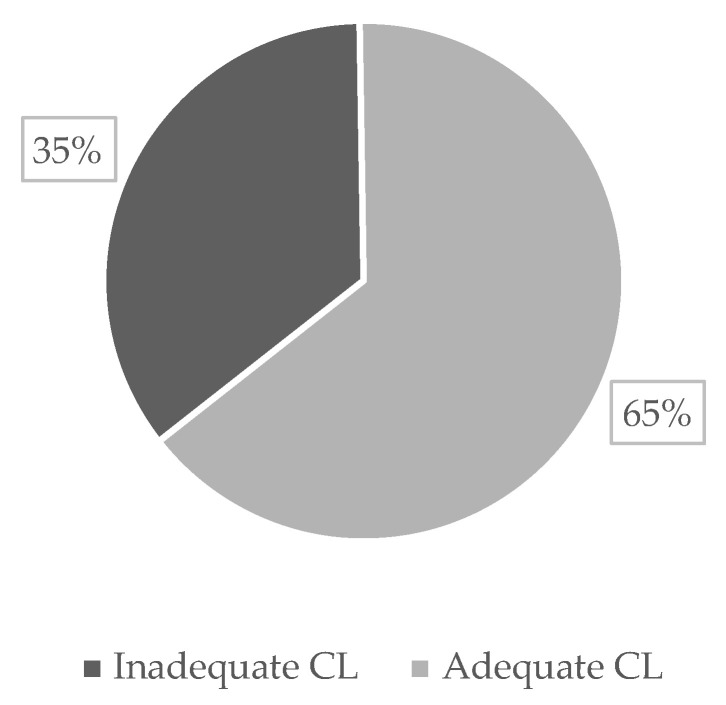
Distribution of CL of the current long-term care recipients (*n* = 652). Data source: A Survey on Older Adults of Sakai City (2019) [20].

**Table 1 ijerph-20-02456-t001:** Details of the setting for the comprehensive index of CL.

Care Literacy (CL)		1: Adequate CL																0: Inadequate CL															
(1) LTCI		1				1				1				0				0				0				1				0			
(2) Nursing Care Facility	SM Care	1	1	1	0	1	1	1	0	0	0	1	0	1	1	1	0	0	0	1	0	1	1	1	0	0	0	1	0	0	0	1	0
	NSM Care	1	1	0	1	1	1	0	1	0	1	0	0	1	1	0	1	0	1	0	0	1	1	0	1	0	1	0	0	0	1	0	0
	RPO Care	1	0	1	1	1	0	1	1	1	0	0	0	1	0	1	1	1	0	0	0	1	0	1	1	1	0	0	0	1	0	0	0
	Sum	3	2			3	2			1			0	3	2			1			0	3	2			1			0	1			0
	Index	1				1				0				1				0				1				0				0			
(3) Future Needs of Care		1				0				1				1				1				0				0				0			
Sum of index (1)–(3)		3				2												1												0			

Note: SM Care stands for “Small-scale multifunctional home care”; NSM Care stands for “Nursing small-scale multifunctional home care (combined service)”, and RPO Care stands for “Regular patrol/on-demand home-visit nursing care”. Data source: A Survey on Older Adults of Sakai City (2019) [20].

**Table 2 ijerph-20-02456-t002:** The two-stage setting of the evaluation of satisfaction with CM.

Aspects	Stage I			
	0: No Evaluation (Either Answered “Not Sure” or Without a Response)	1: With an Evaluation (Either Answered Favorable or Unfavorable)		
(1) Expertise	0	1	Stage II	0: Dissatisfied
				1: Satisfied
(2) Information provision	0	1		0: Dissatisfied
				1: Satisfied
(3) Explanatory power	0	1		0: Dissatisfied
				1: Satisfied
(4) Communication	0	1		0: Dissatisfied
				1: Satisfied
(5) Responsiveness	0	1		0: Dissatisfied
				1: Satisfied
(6) Attitude and manners	0	1		0: Dissatisfied
				1: Satisfied

Data source: A Survey on Older Adults of Sakai City (2019) [20].

**Table 3 ijerph-20-02456-t003:** Distribution of responses to satisfaction with CM from the six aspects of CM measurement.

Aspects	Satisfied	Dissatisfied	“Not Sure” and No Response
(1) Expertise	71.2%	6.2%	22.6%
(2) Information provision	64.2	7.2	28.6
(3) Explanatory power	70.2	5.6	24.2
(4) Communication	60.7	5.0	34.3
(5) Responsiveness	69.1	3.9	27.0
(6) Attitude and manners	78.0	2.7	19.3

Data source: A Survey on Older Adults of Sakai City (2019) [20].

**Table 4 ijerph-20-02456-t004:** Descriptive statistics.

Definition	Obs.	Mean	S.D.	Min	Max
Dependent Variables					
Stage I: Whether or not to evaluate					
(1) Expertise	652	0.744	0.437	0	1
(2) Information provision	652	0.658	0.475	0	1
(3) Explanatory power	652	0.725	0.447	0	1
(4) Communication	652	0.604	0.489	0	1
(5) Responsiveness	652	0.683	0.466	0	1
(6) Attitude and manners	652	0.790	0.408	0	1
Stage II: Satisfaction level					
(1) Expertise	485	0.926	0.262	0	1
(2) Information provision	429	0.890	0.313	0	1
(3) Explanatory power	473	0.930	0.255	0	1
(4) Communication	394	0.924	0.266	0	1
(5) Responsiveness	445	0.944	0.231	0	1
(6) Attitude and manners	515	0.961	0.193	0	1
Main Independent Variable					
Care literacy (CL)	652	0.647	0.478	0	1
Other Covariates					
Age category: 65–69 y	652	0.043	0.203	0	1
70–74 y	652	0.115	0.319	0	1
75–79 y	652	0.227	0.419	0	1
80–84 y	652	0.288	0.453	0	1
85–89 y	652	0.224	0.417	0	1
90+ y	652	0.103	0.304	0	1
Male	652	0.290	0.454	0	1
Family structure: living alone	632	0.397	0.490	0	1
Couple only	632	0.361	0.481	0	1
Other structures	632	0.242	0.429	0	1
Certified degree of care: “Requiring help” levels 1–2	652	0.661	0.474	0	1
”Long-term care” levels 1–2	652	0.256	0.437	0	1
”Long-term care” levels 3–5	652	0.083	0.276	0	1
Reason for choosing the CM’s office (introduced)	592	0.459	0.499	0	1
Utilization status of long-term care services (yes)	652	0.548	0.498	0	1
Respondents answered with others	652	0.150	0.358	0	1

Data source: A Survey on Older Adults of Sakai City (2019) [20].

**Table 5 ijerph-20-02456-t005:** Marginal effects of CL and other covariates on satisfaction with CMs estimated by inverse probability weighting.

Aspects	(1) Expertise				(2) Information Provision				(3) Explanation Power				(4) Communication				(5) Responsiveness				(6) Attitude and Manners			
	Stage I		Stage II		Stage I		Stage II		Stage I		Stage II		Stage I		Stage II		Stage I		Stage II		Stage I		Stage II	
Care literacy (CL)	0.077	**	0.029		0.119	***	0.007		0.114	***	0.050	**	0.141	***	0.039		0.110	***	0.015		0.065	**	0.037	*
65–69 y	0.090		−0.031		0.062		−0.057		0.085		−0.053		0.141		0.006		0.172	***	−0.001		0.075		−0.063	
70–74 y	0.070		−0.049		0.048		−0.029		0.048		0.024		0.092		0.018		0.085		−0.005		0.075	*	−0.036	
75–79 y	0.028		0.018		0.105	*	−0.023		0.071		0.031		0.086		0.037		0.108	*	−0.040		0.074	*	−0.005	
80–84 y	0.035		−0.020		0.010		−0.073		0.008		0.036		0.081		−0.013		0.093		−0.048		0.044		−0.016	
85–89 y	−0.032		−0.001		−0.066		0.019		−0.040		0.054	**	0.056		0.035		0.082		−0.016		−0.004		0.012	
Male	0.061	*	0.044	**	0.040		−0.004		0.033		0.045	**	0.006		0.003		0.003		−0.070	**	−0.015		0.016	
Living alone	0.057		−0.005		0.050		−0.053		0.023		0.022		0.082		−0.013		0.050		−0.139	***	0.051		−0.024	
Couple only	−0.055		−0.010		−0.038		0.005		0.024		0.015		0.001		0.042		0.000		−0.030		0.006		−0.010	
“Long-term care” levels 1–2	−0.034		−0.045		−0.007		−0.041		0.000		0.002		0.101	*	0.008		0.104	**	0.065	***	−0.010		−0.006	
“Long-term care” levels 3–5	0.056		0.004		0.111		−0.053		0.123	**	−0.006		0.188	***	−0.100		0.107	*	−0.009		0.070		−0.003	
Reason for choosing the CM’s office (introduced)	0.014		−0.056	**	0.037		−0.055	*	0.034		−0.059	***	−0.018		−0.056	*	−0.043		−0.010		0.043		−0.009	
Utilization status of long-term care services (yes)	0.088	**	−0.030		0.130	***	0.022		0.066		−0.010		0.088	*	−0.015		0.102	**	−0.036		0.058	*	−0.007	
Respondents answered with others	0.082	**	0.021		0.070		0.024		0.119	***	0.032	*	0.067		0.019		0.091	*	−0.085		0.100	***	0.008	
*n*	577		457		577		409		577		445		577		374		577		423		577		487	
Pseudo R-squared	0.054		0.074		0.064		0.044		0.062		0.135		0.057		0.099		0.067		0.238		0.063		0.070	

Note: *** *p* < 0.01; ** *p* < 0.05; * *p* < 0.1. Data source: A Survey on Older Adults of Sakai City (2019) [20].

**Table 6 ijerph-20-02456-t006:** Marginal effects of CL and other covariates on satisfaction with CMs estimated by Heckman probit model.

Aspects	(1) Expertise				(2) Information Provision				(3) Explanation Power				(4) Communication			(5) Responsiveness				(6) Attitude and Manners		
	Stage I (Selection)		Stage II (Outcome)		Stage I (Selection)		Stage II (Outcome)		Stage I (Selection)		Stage II (Outcome)		Stage I (Selection)		Stage II (Outcome)	Stage I (Selection)		Stage II (Outcome)		Stage I (Selection)		Stage II (Outcome)
Care literacy (CL)	0.075	**	0.030		0.119	***	0.009		0.115	***	0.049	*	0.141	***	0.031	0.111	***	0.026		0.065	*	0.036
65–69 y	0.089		−0.036		0.066		−0.063		0.091		−0.079		0.144		−0.007	0.171	***	0.007		0.076		−0.092
70–74 y	0.074		−0.057		0.052		−0.038		0.056		0.015		0.094		0.004	0.084		−0.012		0.077	*	−0.060
75–79 y	0.027		0.014		0.107	*	−0.021		0.076		0.024		0.088		0.032	0.106	*	0.015		0.075	*	−0.016
80–84 y	0.035		−0.023		0.010		−0.065		0.015		0.025		0.082		−0.007	0.093		−0.032		0.045		−0.029
85–89 y	−0.033		−0.001		−0.067		0.019		−0.034		0.049	**	0.058		0.049	0.081		−0.003		−0.002		0.002
Male	0.060		0.042	**	0.040		−0.002		0.033		0.046	***	0.006		0.017	0.003		−0.012		−0.014		0.017
Living alone	0.058		−0.010		0.051		−0.052		0.020		0.016		0.081		0.010	0.050		−0.078	*	0.051		−0.025
Couple only	−0.054		−0.011		−0.043		0.011		0.020		0.014		−0.001		0.042	0.002		−0.029		0.005		−0.006
“Long-term care” levels 1–2	−0.037		−0.030		−0.011		−0.036		0.000		−0.002		0.101	**	−0.020	0.104	**	0.015		−0.009		−0.011
“Long-term care” levels 3–5	0.052		0.016		0.103		−0.041		0.123	**	−0.009		0.187	***	−0.031	0.109	*	−0.013		0.070		−0.007
Reason for choosing the CM’s office (introduced)	0.014		−0.056	**	0.036		−0.051	*	0.032		−0.055	**	−0.018		−0.030	−0.043		−0.004		0.043		−0.011
Utilization status of long-term care services (yes)	0.094	**	0.007		0.134	***	0.017	**	0.061		0.003		0.088	*	0.004	0.101	**	−0.003		0.058		0.001
Respondents answered with others	0.083	**	0.006	*	0.072		0.008		0.123	***	0.006	**	0.066		0.003	0.091	*	−0.003		0.100	***	0.002
*n*	577		457		577		409		577		445		577		374	577		423		577		487
LR test of indep. Eqns. (rho = 0)	Prob > chi2 = 0.4043				Prob > chi2 = 0.4432				Prob > chi2 = 0.5042				Prob > chi2 = 0.8420			Prob > chi2 = 0.8793				Prob > chi2 = 0.8062		

Note: *** *p* < 0.01; ** *p* < 0.05; * *p* < 0.1. Data source: A Survey on Older Adults of Sakai City (2019) [20].

**Table 7 ijerph-20-02456-t007:** Marginal effects of CL and other covariates on satisfaction with CMs estimated by probit model.

Aspects	(1) Expertise				(2) Information Provision				(3) Explanation Power				(4) Communication			(5) Responsiveness				(6) Attitude and Manners			
	Stage I		Stage II (Independent eq.)		Stage I		Stage II (Independent eq.)		Stage I		Stage II (Independent eq.)		Stage I		Stage II (Independent eq.)	Stage I		Stage II (Independent eq.)		Stage I		Stage II (Independent eq.)	
Care literacy (CL)	0.077	**	0.029		0.119	***	0.008		0.114	***	0.050	**	0.141	***	0.032	0.110	***	0.026		0.065	**	0.038	*
65–69 y	0.090		−0.029		0.062		−0.065		0.085		−0.069		0.141		−0.007	0.172	***	0.006		0.075		−0.088	
70–74 y	0.070		−0.053		0.048		−0.035		0.048		0.018		0.092		0.004	0.085		−0.012		0.075	*	−0.056	
75–79 y	0.028		0.019		0.105	*	−0.021		0.071		0.026		0.086		0.032	0.108	*	0.014		0.074	*	−0.015	
80–84 y	0.035		−0.019		0.010		−0.067		0.008		0.027		0.081		−0.007	0.093		−0.031		0.044		−0.029	
85–89 y	−0.032		0.000		−0.066		0.022		−0.040		0.049	**	0.056		0.050	0.082		−0.004		−0.004		0.003	
Male	0.061	*	0.045	**	0.040		−0.003		0.033		0.047	***	0.006		0.018	0.003		−0.012		−0.015		0.016	
Living alone	0.057		−0.006		0.050		−0.049		0.023		0.025		0.082		0.011	0.050		−0.076	**	0.051		−0.023	
Couple only	−0.055		−0.009		−0.038		0.016		0.024		0.020		0.001		0.042	0.000		−0.028		0.006		−0.006	
“Long-term care” levels 1–2	−0.034		−0.044		−0.007		−0.035		0.000		0.000		0.101	*	−0.019	0.104	**	0.015		−0.010		−0.006	
“Long-term care” levels 3–5	0.056		0.006		0.111		−0.039		0.123	**	−0.055		0.188	***	−0.031	0.107	*	−0.013		0.070		−0.001	
Reason for choosing the CM’s office (introduced)	0.014		−0.055	**	0.037		−0.052	*	0.034		−0.055	***	−0.018		−0.030	−0.043		−0.004		0.043		−0.011	
Utilization status of long-term care services (yes)	0.088	**	0.025		0.130	***	0.011		0.066		−0.011		0.088	*	0.002	0.102	**	−0.003		0.058	*	−0.010	
Respondents answered with others	0.082	**	0.022		0.070		0.030		0.119	***	0.032	*	0.067		0.005	0.091	*	−0.003		0.100	***	0.008	
*n*	577		457		577		409		577		445		577		374	577		423		577		487	
Pseudo R-squared	0.054		0.081		0.064		0.044		0.062		0.137		0.057		0.048	0.067		0.07		0.063		0.081	

Note: *** *p* < 0.01; ** *p* < 0.05; * *p* < 0.1. Data source: A Survey on Older Adults of Sakai City (2019) [20].

## Data Availability

It is possible to obtain the data by applying for information disclosure to the Longevity Support Division of the Longevity Society Department, Health and Welfare Bureau of Sakai City, Osaka Prefecture. Access link: https://www.city.sakai.lg.jp/shisei/gyosei/shishin/fukushi/kourei-kaigo_keikaku/75402820211117160542577.html (In Japanese) (accessed on 26 January 2023).

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
