# Peer review of "Does Long-Term Care Literacy Matter in Evaluating Older Care Recipients’ Satisfaction with Care Managers? Empirical Evidence from Japanese Survey Data"

_ijerph, 2023, doi:10.3390/ijerph20032456_

Round 1

Reviewer 1 Report

Dear Authors, Congratulations on choosing your field of study due to the
importance of scientific work for practical activity. The work was well
organized but needs some adjustments. Consider omitting the word
"non-respondent bies" from your keywords. The introduction presents
forecasts of the aging of the Japanese society until 2025, but there is
no information about the current demographic situation. This would help
illustrate the rate of aging of the population. This needs to be
supplemented. It is also worth including here information about CM's
education and competences. Who and how can become a CM? In the Methods
section, please provide a more detailed description of the methods used.
Some information can be transferred from the attachment. I also propose
to describe here why the Sakei City study can be generalized to all of
Japan. The drawings included in the attachment (Fig. A1) panels A, B, C
can be combined into one clear table. The information provided in panels
D and E appears to be less relevant. The tables in Annexes A2, A3 and A4
can be included in the Results section. As in Figure 1. Discussion is a
commentary on the obtained results. Limitations of the study and
directions for future research were indicated. Appendix B is redundant
as the description of the method should be in the Methods section. The
study was conducted in 2019, so 3 years ago. Only 7 out of 24 items are
cited in the last 5 years. It is worth checking if there are any newer
publications.

Author Response

Reply Letter to Reviewers

Dear Authors, Congratulations on choosing your field of study due to the importance of scientific work for practical activity. The work was well organized but needs some adjustments. Consider omitting the word "non-respondent bies" from your keywords. The introduction presents forecasts of the aging of the Japanese society until 2025, but there is no information about the current demographic situation. This would help illustrate the rate of aging of the population. This needs to be supplemented. It is also worth including here information about CM's education and competences. Who and how can become a CM? In the Methods section, please provide a more detailed description of the methods used. Some information can be transferred from the attachment. I also propose to describe here why the Sakei City study can be generalized to all of Japan. The drawings included in the attachment (Fig. A1) panels A, B, C can be combined into one clear table. The information provided in panels D and E appears to be less relevant. The tables in Annexes A2, A3 and A4 can be included in the Results section. As in Figure 1. Discussion is a commentary on the obtained results. Limitations of the study and directions for future research were indicated. Appendix B is redundant as the description of the method should be in the Methods section. The study was conducted in 2019, so 3 years ago. Only 7 out of 24 items are cited in the last 5 years. It is worth checking if there are any newer publications.

-Response: Thank you very much for your insights and helpful suggestions. We have carefully considered the suggested revisions to improve our manuscript. We would like to reply to your comments one by one, as presented below:

  1. “The work was well organized but needs some adjustments. Consider omitting the word "non-respondent bies" from your keywords.”

-Response: Thank you for this comment. Following your suggestion, we have removed “non-respondent bias” from the keywords and would like to add “satisfaction” as a new keyword. Please refer to page 1, line 32-33 (tracked version).

  1. “The introduction presents forecasts of the aging of the Japanese society until 2025, but there is no information about the current demographic situation. This would help illustrate the rate of aging of the population. This needs to be supplemented.”

-Response: Thank you for this comment. Following your suggestion, we have added Figure 1 to explain the population structure by age group in Japan from 1995 to 2000 and Figure 2 to describe the percentage of people certified as requiring long-term care among older adults from 2000 to 2020. Please refer to pages 1-3, lines 41-66, 78-95, Figures 1 and 2 (tracked version).

  1. “It is also worth including here information about CM's education and competences. Who and how can become a CM? “

-Response: Thank you for this comment. Following your suggestion, we have added two paragraphs to provide more information on CM’s education and qualifications, and we also pointed out existing problems. We further added Figure 3 and Figure 4 to show specific workflows of CM. Please refer to pages 3-5, lines 97-98, 100-105, 110-130, 134-140 Figures 3 and 4 (tracked version).

  1. In the Methods section, please provide a more detailed description of the methods used. Some information can be transferred from the attachment.”

-Response: Thank you for this comment. Following your suggestion, we have transferred related information from the supplementary information to the main text for a more detailed description of the methods used. Please refer to pages 11-12, lines 302-311, 325-360 (tracked version).

  1. “I also propose to describe here why the Sakei City study can be generalized to all of The drawings included in the attachment (Fig. A1) panels A, B, C can be combined into one clear table. The information provided in panels D and E appears to be less relevant.”

-Response: Thank you for this comment. In the main text, we explained why we could use Sakai City’s data to represent the whole country (page 6-7, lines 181-194 (tracked version)). Following your suggestion, we have moved the figure from the supplementary information and combined several figures into one. We would like to keep panels D and E because we consider that education level and academic ability are related to literacy. A renewed figure of old panels is now shown in Figure 5 in the revised manuscript. Please refer to page 6-7, lines 185-194, Figure 5 (tracked version).

  1. “The tables in Annexes A2, A3 and A4 can be included in the Results section. As in Figure 1. Appendix B is redundant as the description of the method should be in the Methods section.”

-Response: Thank you for this comment. We have moved Tables A2 and A3 from the supplementary information to the Methods section of the main text to better describe how we generated the CL and CM measures. We have also moved Table A4 to this section to explain why we use this framework. Please refer to page 7-8, lines 229-230, Table 1; pages 9-10, lines 264-265, Table 2; pages 10-11, lines 288-290, Table 3 (tracked version).

  1. “The study was conducted in 2019, so 3 years ago. Only 7 out of 24 items are cited in the last 5 years. It is worth checking if there are any newer publications.”

-Response: Thank you for this comment. Following your suggestions, we have further added 10 latest references related to our study. Please refer to References 2-5, 9-13, 34 (tracked version).

Reviewer 2 Report

See attached file. 

Author Response

Reply Letter to Reviewers

The topic of health literacy and its impact on health care access, quality, and outcomes is of great importance, and this article is based on a robust data set to answer questions about those connections. The specific focus on health literacy and its implications specifically for long-term care quality begins to fill a major gap. However, there are few issues that should be addressed before the article can achieve its full potential. Throughout the article, there needs to be greater clarity about the specific research questions, IVs, and DVs, and better attention to the big picture. Here are some specific examples and suggestions:

-Response: Thank you very much for your insights and helpful suggestions. We have carefully considered the suggested revisions to improve our manuscript. We would like to reply to your comments one by one, as presented below:

  1. The article addresses two main topics: whether long-term care literacy (CL) influences an individual’s likelihood of completing an evaluation of case management (CM) services received, and whether CL influences satisfaction ratings on a number of CM quality items. Using the term “behavior” for whether or not someone answers questions about CM is confusing, as is the mention of “ability to evaluate satisfaction”. You are really only talking about willingness to complete all of the questions on the survey, correct? This needs to be clearer.

-Response: Thank you for this comment, and we apologize for the confusion. In this study, we focused on the behavior of older adults and whether or not they answered the question regarding the satisfaction of CM. We have revised the related description in the Abstract and Conclusion. Please refer to page 1, lines 13-17; page23, lines 473-481; page 24, lines 531-533 (tracked version).

  1. The results do not clearly answer the two big questions (CL impact on completing a CM evaluation, CL impact on ratings given), and the discussion seems to focus on demographic factors that influence satisfaction ratings, and on specific aspects of CM that should be improved. There is little about the importance of CL, which is the prmise on which the article is based.

-Response: Thank you for this comment. Following your suggestion, we have expanded the discussion on the impacts of CL in the first paragraph of the Discussion section. We have also summarized the impacts of CL on evaluation and rating in the first paragraph of the Conclusion section; furthermore, in the second paragraph, we have emphasized the importance of improving CL. Please refer to page 23, lines 473-481 (tracked version).

  1. The focus on responses to the specific CM items draws the reader into a details, but we lose the big picture. Help us understand the bigger picture, and give us more depth about the concepts related to CM roles and quality that the specific items are intended to measure.

-Response: Thank you for this comment. Following your suggestion, we have added Figures 3 and 4 to show specific workflows of CM, showing why we focused on these specific CM items. Please refer to pages 3-5, lines 97-98, 100-105, 110-130, 134-140 Figures 3 and 4 (tracked version).

I have the following additional suggestions to improve the utility and readability of the

paper:

  1. Readers need a bit more background about the CM system in Japan. Are CMs employed by health systems, a governmental agency, private agencies specializing in this function, all of the above? How do people typically get linked to the services provides by CMs? What proportion of people typically use CM services to meet their long-term care needs.

-Response: Thank you for this comment. Following your suggestion, we have added several paragraphs to provide more information on the role, education, and qualifications of CMs in the Introduction, and we have also highlighted existing problems to emphasize the importance of studying CM (page 4, lines 100-105; page 5, lines 134-140 (tracked version)).

CMs are employed by an in-home care support office. To carry out an in-home nursing care support business, it is necessary to have a corporate status (such as a stock company, limited liability company, NPO corporation, or social welfare corporation), apply for approval after meeting the standards set by the MHLW, and receive the designation of the office from the municipality.

Anyone in need of long-term care and desiring to receive formal care services must be certified as needing long-term care and consult their needs with a CM who can be introduced by public institutions, referred by word of mouth, or found by themselves. The CM creates a care plan, and care services are started based on that plan.

Approximately 20% of those certified for long-term care do not use the formal care service, and it is found that people with mild care needs do not use it frequently [1(in Japanese)].

References:

[1] Kaibara, R., Ueno, M. & Izumi, K. (2016) The Relation between Unusing Long-term Care Insurance Service and Social Interaction in Elderly People with Lower Care Levels. Nursing Journal of Mukogawa Women’s University vol.01, 29-36

http://doi.org/10.14993/00000791

  1. It is unclear how the Appendix panels A through E are important to the article.

-Response: Thank you for this comment. We use these figures to show that the demographic and socio-economic characteristics (level of income, education, academic ability) of Sakai City and Japan as a whole were consistent. This is important to show that Sakai City’s data could represent the entire country. Following your suggestion, we have moved the figure from the supplementary information and combined several figures into one. A renewed figure of old panels is now shown in Figure 5 in the revised manuscript. Please refer to page 6-7, lines 185-194, Figure 5 (tracked version).

Reviewer 3 Report

Thank you for the opportunity to review this manuscript. Authors did a good job discussing the needs for the study and explaining the methods section. The analyses were thoroughly explained as well. I do see its potential for publication, but I have a few questions which will be outlined below, for the authors to consider.

1. I would suggest providing some information about how care managers function in Japan. That is, how are they selected if it is not through referral and how they become “licensed caregivers” (page 2, line 62), and if each older adult in Japan who is qualified for long-term care will be assigned a care manager. Some explanations of CM will help your readers understand the context better.

2. This is a very minor point – whenever the “six aspects” are mentioned in the paper, make sure it is written as “the six aspects of CM measurement” so your readers know what it is referred to.

3. There are a number of data collapsing here in the analyses. I understand the needs to do so statistically. However, I would like to suggest authors considering potential impacts on the results and also conclusions drawn from these analyses.

4. On pages 9 and 10, authors reported the results by age group breakdown. I am not sure the meaning of such differentiation. In the discussion, there is barely any mentioning of these different analyses, except for gender difference. I like to know more about the purpose of this part of the analyses in the revision, if granted.

4. I would suggest authors avoid using the terms, such as “effect” or “impact on” in the paper, especially when it comes to explaining the results. This is a cross-sectional study, which means the causal relationship is un-determined. Relatedly, in the discussion authors suggested that the politer “explanation power” and “attitude and manners” of CM might be required (page 16, lines 317-318, and again on page 17, line 362-363). If the results are correlational, we can also say a recommendation of this result could be to increase care recipient’s CL. This may allow them to engage in more meaningful conversations and discussions with their CM, which in turn, the ability to evaluate their CM’s performance. What I am trying to say is that since it is correlational, we should discussion the results from both directions. The latter is mentioned in the rationale for the hypothesis (page 2, lines 78-83) but when it comes to discussing the results, the opposite was suggested. The additional discussion may strengthen the depth of the discussion.

5. I Would like to suggest adding some information at the front end about how older adults acquire CL in Japan. This addition may help addressing the practical implications of the results in the discussion.

Author Response

Reply Letter to Reviewers

Thank you for the opportunity to review this manuscript. Authors did a good job discussing the needs for the study and explaining the methods section. The analyses were thoroughly explained as well. I do see its potential for publication, but I have a few questions which will be outlined below, for the authors to consider.

-Response: Thank you very much for your insights and helpful suggestions. We have carefully considered the suggested revisions to improve our manuscript. We would like to reply to your comments one by one, as presented below:

  1. I would suggest providing some information about how care managers function in Japan. That is, how are they selected if it is not through referral and how they become “licensed caregivers” (page 2, line 62), and if each older adult in Japan who is qualified for long-term care will be assigned a care manager. Some explanations of CM will help your readers understand the context better.

-Response: Thank you for this comment. Following your suggestion, we have added several paragraphs to provide more information on the role, education, and qualifications of CMs, and we have also highlighted existing problems. We have further added Figures 3 and 4 to show specific workflows of CM (pages 3-5, lines 97-98, 100-105, 110-130, 134-140 Figures 3 and 4 (tracked version).).

CMs are employed by an in-home care support office. To carry out an in-home nursing care support business, it is necessary to have a corporate status (such as a stock company, limited liability company, NPO corporation, or social welfare corporation), apply for approval after meeting the standards set by the MHLW, and receive the designation of the office from the municipality.

As you mentioned, each older adult in Japan who is qualified for long-term care requires a CM to help them to generate a care plan to receive formal care services. A CM can be introduced by public institutions such as the Regional Comprehensive Support Center and Home Care Station, or referral by word of mouth such as acquaintance or friend, or found by themselves. The CM creates a care plan, and care services are started based on that plan.

  1. This is a very minor point – whenever the “six aspects” are mentioned in the paper, make sure it is written as “the six aspects of CM measurement” so your readers know what it is referred to.

-Response: Thank you for this comment. Following your suggestion, we have revised “six aspects” to “the six aspects of CM measurement.” Please refer to page 1, line 14; page 10, line 282, 288; page 16, lines 381-382, 417; page 23, line 474; page 24, line 532 (tracked version).

  1. There are a number of data collapsing here in the analyses. I understand the needs to do so statistically. However, I would like to suggest authors considering potential impacts on the results and also conclusions drawn from these analyses.

-Response: Thank you for this comment. In the Introduction section, we hypothesize the path of potential impacts of CL, and based on the results, we have provided a further explanation in the Discussion section. Please refer to page 6, lines 154-161; page 23, lines 473-481 (tracked version).

  1. On pages 9 and 10, authors reported the results by age group breakdown. I am not sure the meaning of such differentiation. In the discussion, there is barely any mentioning of these different analyses, except for gender difference. I like to know more about the purpose of this part of the analyses in the revision, if granted.

-Response: Thank you for this comment. In the Discussion section, we have added a paragraph to discuss the difference in results by different age groups (page 23, lines 490-498 (tracked version)). It might be better to use real age data as a covariate, but this information is not accessible because the dataset is secondary and age was classified in 5-year increments. We believe that this can be improved in future studies.

  1. I would suggest authors avoid using the terms, such as “effect” or “impact on” in the paper, especially when it comes to explaining the results. This is a cross-sectional study, which means the causal relationship is un-determined. Relatedly, in the discussion authors suggested that the politer “explanation power” and “attitude and manners” of CM might be required (page 16, lines 317-318, and again on page 17, line 362-363). If the results are correlational, we can also say a recommendation of this result could be to increase care recipient’s CL. This may allow them to engage in more meaningful conversations and discussions with their CM, which in turn, the ability to evaluate their CM’s performance. What I am trying to say is that since it is correlational, we should discussion the results from both directions. The latter is mentioned in the rationale for the hypothesis (page 2, lines 78-83) but when it comes to discussing the results, the opposite was suggested. The additional discussion may strengthen the depth of the discussion.

-Response: Thank you for this comment. We agree with your opinion that the endogeneity problem might still exist even though we attempted to control it by applying several econometric models. We have mentioned it as a limitation of this study in the Discussion section (page 24, lines 523-527 (tracked version)). Following your suggestion, we have avoided using terms such as “effect” and “impact on” when summarizing our results throughout the paper.

  1. I Would like to suggest adding some information at the front end about how older adults acquire CL in Japan. This addition may help addressing the practical implications of the results in the discussion.

-Response: Thank you for this comment. Following your suggestion, we have added further information regarding how older adults require information related to health care and nursing care and highlighted existing problems that may lead to low CL. Owing to the dearth of studies on CL in Japan and even globally, we cannot add more references of CL, but we would like to emphasize the importance of obtaining sufficient information that may lead to improved literacy and improve the CL of older adults. Please refer to pages 1-2, lines 41-63; page 24, lines 546-550, 558-561 (tracked version).
